# Long-Term Environmental Monitoring in an Arctic Lake Polluted by Metals under Climate Change

**Elena M. Zubova [1,\*], Nikolay A. Kashulin [1,\*], Vladimir A. Dauvalter [1] 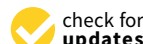, Dmitry B. Denisov [1], Svetlana A. Valkova [1], Oksana I. Vandysh [1], Zakhar I. Slukovskii [1,2], Peter M. Terentyev [1] and Alexander A. Cherepanov [1]**

1   Institute of North Industrial Ecology Problems of Kola Science Center of the Russian Academy of Sciences, 184209 Apatity, Russia; vladimir@dauvalter.com (V.A.D.); proffessuir@gmail.com (D.B.D.); valkovas878@gmail.com (S.A.V.); o.vandysh@ksc.ru (O.I.V.); slukovsky87@gmail.com (Z.I.S.); pterentjev@mail.ru (P.M.T.); acher05503@gmail.com (A.A.C.)
2   Institute of Geology of Karelian Research Centre of the Russian Academy of Sciences, 185910 Petrozavodsk, Russia
\*   Correspondence: seelewolf84@yandex.ru (E.M.Z.); kashulin@mail.ru (N.A.K.)

**Abstract:** Lake Kuetsjarvi (in the lower reaches of the Pasvik River, Murmansk Region, Russia) in the border area between Russia and Norway, is one of the most polluted water reservoirs in the European Arctic. The operation of the Pechenganikel Smelter located on its shores has led to the extremely high concentrations of heavy metals observed in the waters and sediments of the lake. Long-term comprehensive studies of the ecosystem of Lake Kuetsjarvi have made it possible to identify the response of its components to the global and regional change in the environment and climate as a whole, resulting in increased water toxicity and eutrophication, reduction in the number of stenobiont species of aquatic organisms against the background of an increase in the number of eurybiontic and invasive species. Modern communities of Lake Kuetsjarvi are the result of a combination of long-term changes in the abiotic environment and biotic interactions. Heavy-metal pollution of Lake Kuetsjarvi, observed since the 1930s, has led to the formation of a community that is resistant to this type of impact and supports large populations of adapted species. Adaptations of communities to the dynamics of the environmental conditions that their members are exposed to include changes in the species composition, quantitative indicators, ratios between individual taxonomic groups, and the population structure. The development of sympatric forms that differ in the ecological niches they occupy, morphology, and life cycle strategies, including the transition to a short-cycle survival strategy, allows whitefish to remain the dominant species and maintain high population numbers. Unlike the organismal level, responses to medium-term environmental changes on the population and community level are less specific and characterized by stronger inertia.

**Keywords:** pollution; heavy metals; ecosystem; Lake Kuetsjarvi

## 1. Introduction

Being one of the key natural resources of the Arctic, water bodies have undergone major changes caused by the intensive industrial development in the region. The high level of industrial development in the European Arctic has led to radical changes in the structural and functional organization of natural ecosystems, reducing their resource potential [1–7]. In many of the region's lakes exposed to long-term and intensive industrial pollution, against the background of a less stable regional climate, primary production processes, complex interspecific and symbiotic interactions between aquatic organisms have been disrupted and a change in the species composition of the communities

is observed [8–11]. This demonstrates the relevance of studying the patterns of water reservoir functioning in a dynamic system of interrelated natural and anthropogenic factors as well as identifying the controlling environmental factors, which will allow predicting the future development of ecosystems. Understanding the mechanisms of functioning and transformation of complex natural systems in the context of long-term changes in environmental factors forms the scientific basis for maintaining sustainability, rational management, and protection of water resources.

This paper presents the findings of studies into the dynamics of various ecosystem components of the small subarctic Lake Kuetsjarvi in the period between 1989 and 2016. Lake Kuetsjarvi is part of the Inari (Inarijarvi)–Pasvik (Patsojoki, Paz) river and lake system (the Barents Sea basin, catchment area 18,300 km$^2$). This is a unique water reservoir hosting a population of Europe's smallest whitefish species *Coregonus lavaretus* (Linnaeus, 1758) [12,13], which has for a long time maintained a large population, despite the extremely high levels of heavy metal pollution in its waters [12,14,15]. The main sources of anthropogenic impact on the lake are the industrial sites (smelter, mines, slag storage, related infrastructure) of the Pechenganickel Mine and Smelter located on the shores of the lake and founded in the 1930s by the Finnish company Petsamon Nikkeli. Toxic and acid compounds leached from the waste dumps and rocks as a result of acid rain and meltwater also enter the lake. Dust emissions from the site, and dusting from its waste dumps and outdoor areas have a great impact on the chemical composition of the water and sediments in the lake. Climate change estimates based on data from the Murmansk Department for Hydrometeorology and Environment monitoring network in the study area for the period 1976–2012 show a steady trend of increasing rainfall by between 1.8 mm/month (Janiskoski) and 2.4 mm/month (Nickel) over 10 years and an increase in the average annual temperature by 0.6 °C over 10 years [7].

The research goal of this study was to identify the main patterns of adaptation of the ecosystem of Lake Kuetsjarvi supporting its functioning in extreme environmental conditions.

## 2. Materials and Methods

Description of the lake. Lake Kuetsjarvi is part of the lake and river system of the border Pasvik River, with which it is connected by a small channel (Figure 1). It is one of the largest in the border area (water surface area 17.0 km$^2$, maximum depth 37 m), an elongated glacial lake 11.6 km long and 2.8 km across at its widest. The water exchange rate is 1.55. Morphologically, the catchment area can be described as a combination of gentle depressions of glaciolacustrine plains and denudation and tectonic denudation blocks with an intermittent Quaternary sedimentary cover with elevations up to 631.0 m (Mount Kuorpukas). The lake shores are high, covered with alpine and anthropogenic wastelands, partially covered with mixed forest. Coniferous forests are largely degraded. Due to the deposition of acid emissions from the smelter, blasting operations, slag dumps, and quarry sites spread across a vast territory, the natural landscape is significantly modified with its mountain peaks eroded and orography modified. In terms of water quality, the lake is one of the most polluted in Northern Europe [7,16]. Through the Kolosjoki River, the lake receives wastewater from the Pechenganickel Smelter (mine waters, runoff from slag storage sites). Municipal sewage from the urban community of Nickel also enters the lake. Surface runoff carries with it heavy metals deposited in the catchment area into the lake. Their mobility in the natural environment increases as a result of anthropogenic acidification [17]. The main pollutants are Ni, Cu, and Co, as well as their associated elements Pb, As, Cd, and Hg.

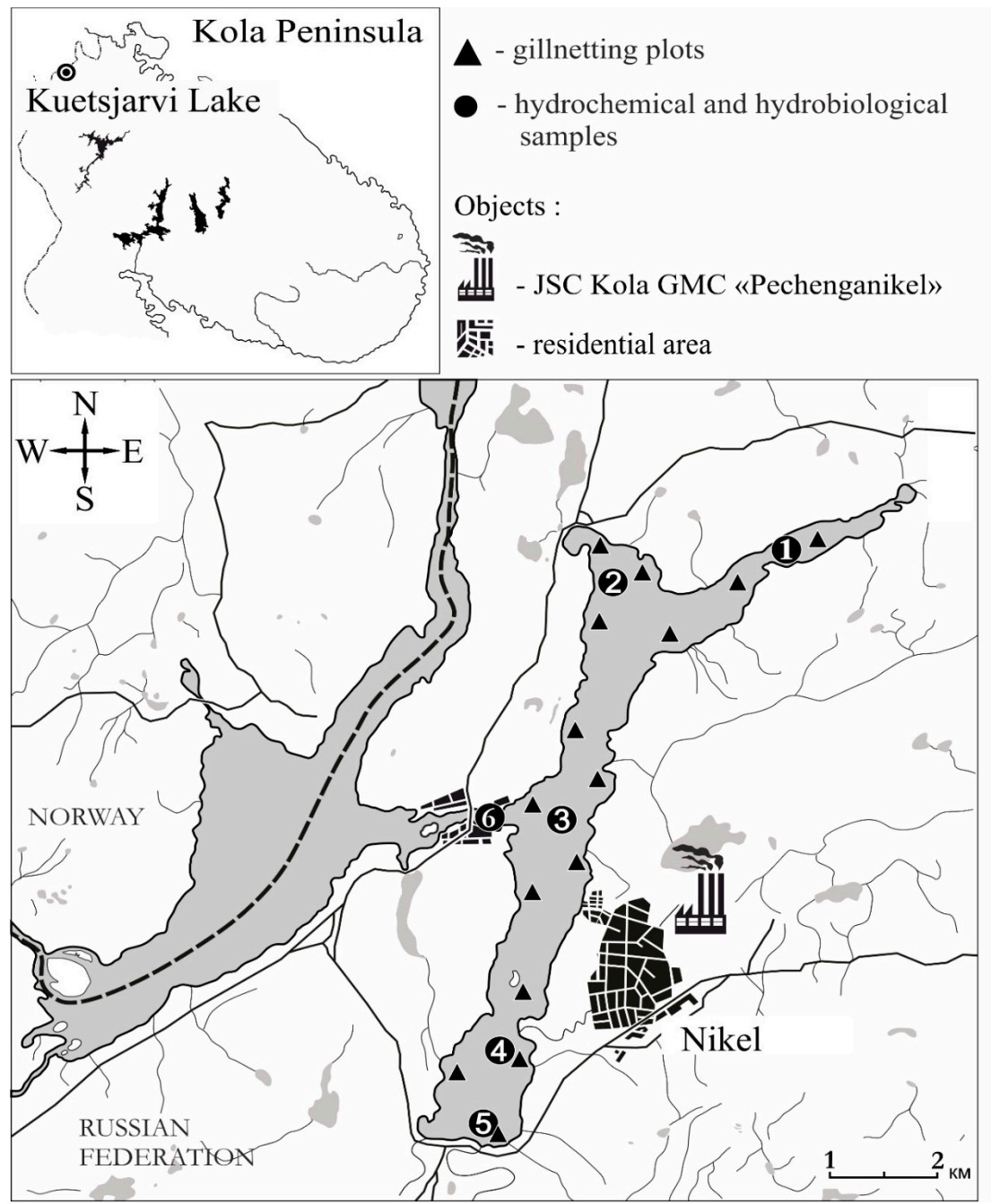

**Figure 1.** Map outline of Lake Kuetsjarvi.

Chemical sampling. This paper presents the hydrochemical monitoring data of Lake Kuetsjarvi from 1989 to 2016 from a monitoring station located on the channel connecting the lake with the lake and river system of the Pasvik River (Figure 1), as well as the chemical composition data of the sediment core sampled in the deepest part of Lake Kuetsjarvi—at the Bely Kamen station (depth 32 m) (Figure 1). Water samples from the surface layer (1 m from the surface) and the bottom layer (1 m from the bottom) of the lake were collected using a two-liter plastic bathometer. Water chemistry was studied at the Shared Resource Center, Institute of Industrial Ecology of the North, Kola Research Center of the Russian Academy of Sciences, by following standardized methods [5,18]. Sediment core sampling, sample preparation, and chemical analysis methods have been described previously [19]. The environmental condition of the freshwater system was evaluated according to the

method proposed by L. Håkonson [20]. The measurement accuracy of the level of chemical elements was controlled by comparing with the analysis results of the reference sample L6M (sediment sample, Finnish Environment Institute (SYKE) 06/2008) and by taking part in comparative tests as part of an international intercalibration program [21]. Correlation and regression analyzes were used to interpret hydrochemical parameters using the program Excel 2003. Graphs of the vertical distribution of metal concentrations in lake sediments are also constructed using Excel 2003.

The shape and composition of the mineral particles in the sediments were studied using a VEGA II LSH scanning electron microscope (SEM) with an INCA Energy 350 energy dispersive microanalyzer at the Analytical Center of the Institute of Geology, Karelian Research Center of the Russian Academy of Sciences in Petrozavodsk. Particles were studied using backscattered electron and secondary electron microscopy and quantitative chemical analysis obtained using energy dispersive X-ray spectrometry.

Hydrobiological studies. Sampling and sample analysis of phytoplankton were carried out annually, in late July or early August in the period 2007–2015, of zooplankton at the same time in the period 1993–2015 according to Russian standard 17.1.3.07–82 [22] (Figure 1), using the recommended standardized methods [23–26] as described earlier [27,28]. In 2012 and 2013, plankton samples were collected during all summer months and in September to assess plankton dynamics during the hydrobiological summer. A total of 112 samples of phytoplankton and 120 samples of zooplankton were collected. The taxonomy of algae and cyanoprokaryotes is harmonized with the international algological database [29]. Based on the taxonomic composition of phytoplankton, water quality was assessed (including identification of class) based on the saprobity index (S) using the Pantle and Bukk method as modified by Sladechek [30,31]. The environmental characteristics of the identified taxa are based on [32]. The concentrations of photosynthetic pigments were calculated using standard methods generally accepted both globally and in Russia [33]. Trophic status assessment by the content of chlorophyll a and the level of phyto- and zoo-plankton biomass was carried out using the scale proposed by Kitaev [34]. To assess the condition of the phytoplankton communities in Lake Kuetsjarvi in 1994–1998, the data collected by Sharov was used [35,36]. The periods of the lake ecosystem development based on plankton indicators were revealed according to several criteria: (1) change in the species composition—large taxonomic categories proportion; (2) change of the dominant species; (3) change in median abundance along with the growth of cases with extremely high values.

Benthic samples were collected using an Ekman grab (covering an area 1/40 m$^2$) in the profundal area (10–32 m). Quantitative and qualitative sampling in the littoral zone (depth <1 m) was carried out using a hand net (mesh size 0.5 mm). The samples were washed through a sieve with a 0.25-mm mesh size. All animals were picked out and were fixed with a 4% formalin solution or a 70% ethyl alcohol solution. A total of 22 samples were taken. Benthic samples were analyzed using standard methods recommended by the Russian State's standards and Guidance on Methods [22,25]. Animals were preserved in the field in 70% ethanol and sorted and identified later in the laboratory under a stereomicroscope. Invertebrate species were identified using the Identification Guide to Zooplankton and Freshwater Zoobenthos of European Russia [37], Identification Guide to Freshwater Invertebrates of Russia and Adjacent Territories, ed. by Tsalolikhin [38,39], and An Introduction to the Aquatic Insects of North America" [40].

Ichthyological studies. Fish were sampled using standard sets of stationary gill nets of monofilament. In the littoral zone (at a depth of 1.5–3 m), 25 m long nets 1.5 m high with a mesh size of 10–60 mm (to capture fish ≥ 5 cm long) were set. The nets were set in groups of 1–2 perpendicular to the shore on sites with sand and gravel banks and large boulder debris. In the profundal zone with depths of more than 18 m, up to 10 multiple mesh nets were set in the same group. To sample ichthyofauna in the pelagic zone, floating multisize nets 3 m high were used. Ichthyofauna sampling sites are shown in Figure 1. Between 1990 and 2015, 3121 specimens of different species of fish were examined. Detailed information on the sample size and catch time is presented in Table 1.

**Table 1.** Characteristics of the used ichthyological material from Lake Kuetsjarvi, 1990–2015.

| Research Period | The Ratio of the Number of Fish Species in Catches, Specimens: | | | | | | |
| --- | --- | --- | --- | --- | --- | --- | --- |
| | Trout | Whitefish | Vendace | Grayling | Pike | Perch | Burbot |
| August 1990 | 6 | 150 | - | 1 | 51 | 45 | 1 |
| September 1991 | - | 371 | - | - | 28 | 40 | - |
| June–September 1992 | - | 225 | - | - | - | - | - |
| June–September 1998 | - | 373 | - | - | 2 | 5 | 7 |
| August 2004 | - | 468 | 37 | 1 | 12 | 54 | - |
| August 2005 | 1 | 23 | - | - | - | 17 | - |
| August 2007 | 2 | 177 | - | - | 1 | 57 | - |
| September 2009 | - | 113 | 1 | - | - | 1 | 6 |
| July–August 2012 | 5 | 363 | 23 | - | 12 | 33 | 7 |
| July–September 2013 | - | 90 | 5 | - | 7 | 13 | - |
| September 2015 | 2 | 201 | 34 | - | 2 | 45 | 2 |

Note: '-' denotes an absence in the sample.

The samples were pretreated according to the method proposed by Sidorov and Reshetnikov [41]. Fish body weight was measured to 1 g, fork length (further *FL*) to 1 mm. In the 1990–1998 catches of perch *Perca fluviatilis* Linnaeus, 1758, only fishing length was measured, therefore the linear size measurements of perch from this period are not given. To identify the different forms of the same whitefish species, gill rakers were counted on the first branchial arch. We classified fish individuals as spawning if their gonads were in the following sexual maturity stages: in whitefish and vendace *C. albula* (Linnaeus, 1758) III–IV [42], in perch and pike *Esox lucius* Linnaeus, 1758 II–III (in autumn catches) and III–IV. For the characteristics of each sample of the analyzed fish species calculated the average value and its error. A comparison of linear and weight whitefish characteristics of different ages was performed using the T-test.

## 3. Results and Discussion

Lake hydrochemistry. The waters of Lake Kuetsjarvi are neutral (pH 6.86–7.48). At the beginning of the summer season, the pH values are lower due to the influx of more acid meltwater from the catchment area (Figure 2a). $SO_4^{2-}$ and $Ca^{2+}$ ions dominate in the waters of Lake Kuetsjarvi, although surface waters in Murmansk Region's unpolluted areas typically belong to the hydrocarbon class and the calcium or sodium group [43]. Sulfates average 59% of the lake's overall anionic composition, calcium averages 56% of its cationic composition. Over the 30-year hydrochemical monitoring period, sum of water-soluble ions in Lake Kuetsjarvi reached 80 mg/L (Figure 2b), which is approximately four times the median value characteristic of the natural oligotrophic surface waters of Murmansk Region [44]. The lowest values were observed in the spring–summer period in the southern part of the lake near the mouths of the rivers Shuonijoki and Kolosjoki due to the inflow of low-mineralized meltwater.

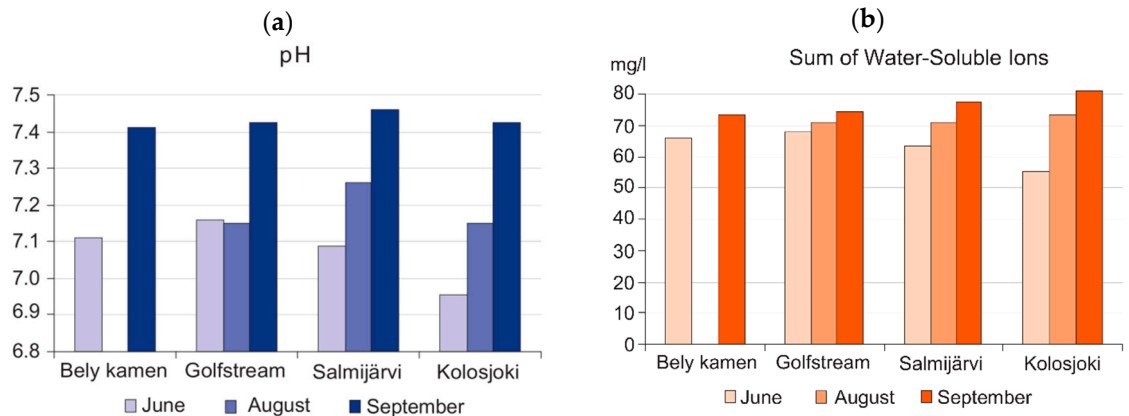

**Figure 2.** Seasonal changes in pH (**a**) and sum of water-soluble ions (**b**), mg/L in Lake Kuetsjarvi, 2013.

Total phosphorus in Lake Kuetsjarvi ranges 11 to 37 µgP/L, averaging 17 µgP/L, total nitrogen 156 to 337 µgN/L, averaging 237 µgN/L. Chemical oxygen demand in Lake Kuetsjarvi ranges from 3.78 to 4.62 mg/L, the water color is quite low in spring (19–26 Pt), summer (15–18 Pt), and autumn (16–17 Pt) seasons. Ni level in Lake Kuetsjarvi varies in the range of 110 to 161 µg/L, averaging 133 µg/L (Figure 3a), Cu 10.4 to 22.0 µg/L, averaging 14.5 µg/L (Figure 3b). Cd, Pb, and Co levels in the lake are close to the detection limit and average 0.1–0.9 µg/L.

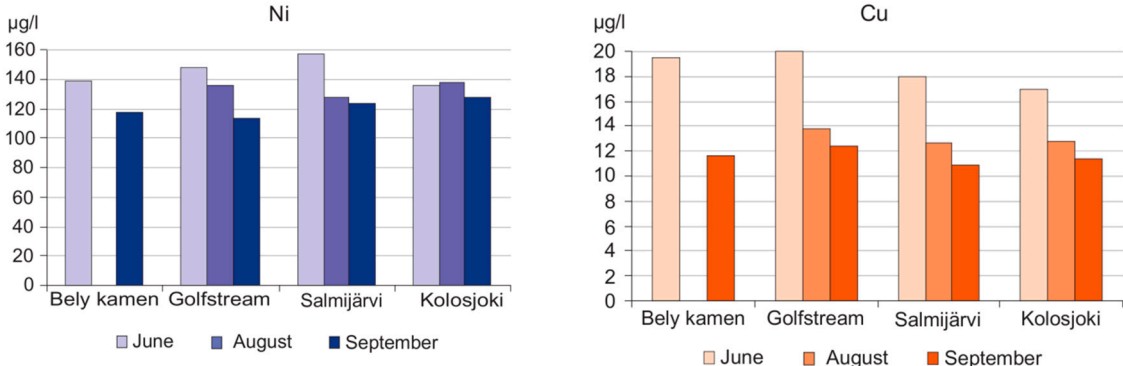

**Figure 3.** Seasonal changes in the contents of Ni (**a**) and Cu (**b**), µg/L in Lake Kuetsjarvi, 2013.

Over the past 30 years, in the context of ongoing pollution, a decrease has been observed in the mineralization of the waters of Lake Kuetsjarvi as a result of a significant ($p < 0.01$) decrease in the content of the basic ions $SO_4^{2-}$, $Cl^-$, $Na^+$, and $K^+$, while the level of toxic HMs (Ni and Cu) has increased and exceeds the background levels in the surface waters of Murmansk Region [44] by tens and hundreds of times (Ni by more than 200 times, Cu by more than 20 times). $SO_4^{2-}$ levels in the lake waters are more than two times higher than the level of the second prevailing $HCO_3^-$ anion in terms of equivalent concentration.

In a study by the Hydrochemical Institute conducted in 1969–1971, close average levels of concentrations of most elements were found in Lake Kuetsjarvi in different parts of the water area and water horizons [45]. A similar trend is observed presently. The level of the lake's major heavy metal pollutant Ni in the southern part of the lake (near the point where the smelter's effluents enter the lake) is only 15% higher than in its northern part.

A comparison of the data on the total concentrations of Ni and Cu showed that over half a century, there has been a significant increase in their levels (Figure 4). The level of not only priority pollutant metals is increasing, but also of Fe and Mn, which are found in the ore fed to the smelter.

There has also been a significant ($p < 0.01$) increase in the past 30 years in the indicators of the level of organic material in the lake waters—chemical oxygen demand ($COD_{Mn}$) and total organic carbon (TOC) (Figure 4), which indicates the intensification of the eutrophication processes in the lake. That said, the sediments in Lake Kuetsjarvi are characterized by a not very high content of organic material—LOI (loss on ignition) in the surface layer reaches 20%. The dominant pollutants are Ni, Cu, Zn, and Co, as well as their associated chalcophilic elements Pb, As, Cd, and Hg (Figure 5). The most polluted are the top 5–10 cm of the lake sediments. Elements such as Cu, Co, Pb, and Cd reach maximum concentrations in the surface layers of the sediment in Lake Kuetsjarvi. The other HMs studied (Ni, Zn, As, and Hg) have maximum concentrations at a depth of 2−5 cm in the sediments (Figure 5). The decrease in the concentration of these elements in the top 1−4 cm of the sediments in Lake Kuetsjarvi can be explained by changes in physicochemical conditions in the lake itself and its catchment area, as well as a decrease in the HM discharge and emissions from the Pechenganickel Smelter. In 1990–2007, Ni discharge dropped from 12.9 to 4.4 tons annually, while Ni emissions into the atmosphere during the same period remain approximately unchanged at 300–350 tons annually http://www.kolagmk.ru/.

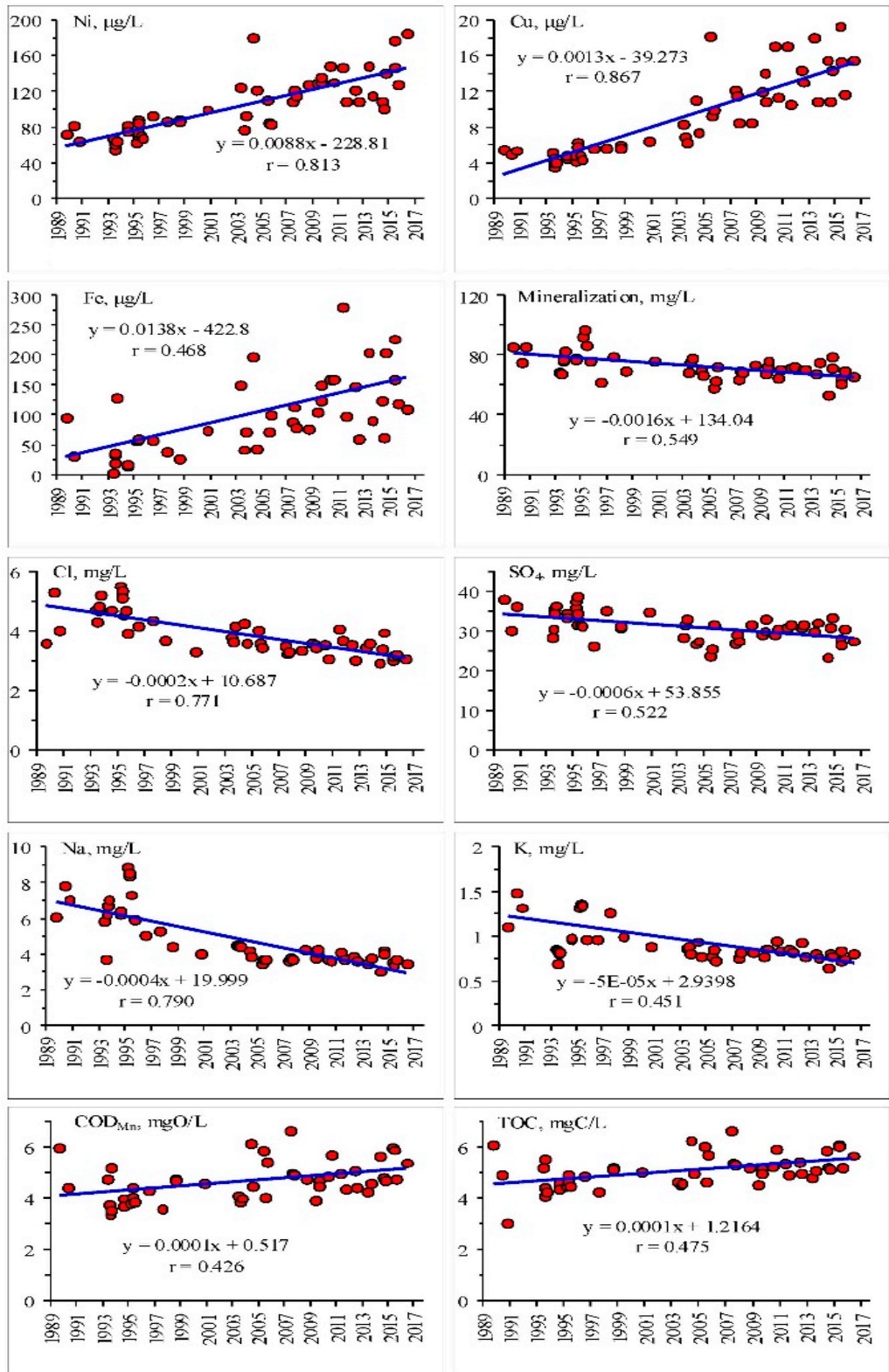

**Figure 4.** The dynamics of hydrochemical indicators in the Lake Kuetsjarvi for the period from 1989 to 2016. The dependences are significant at *r* > 0.354 (*p* < 0.01) with a sample of n = 52.

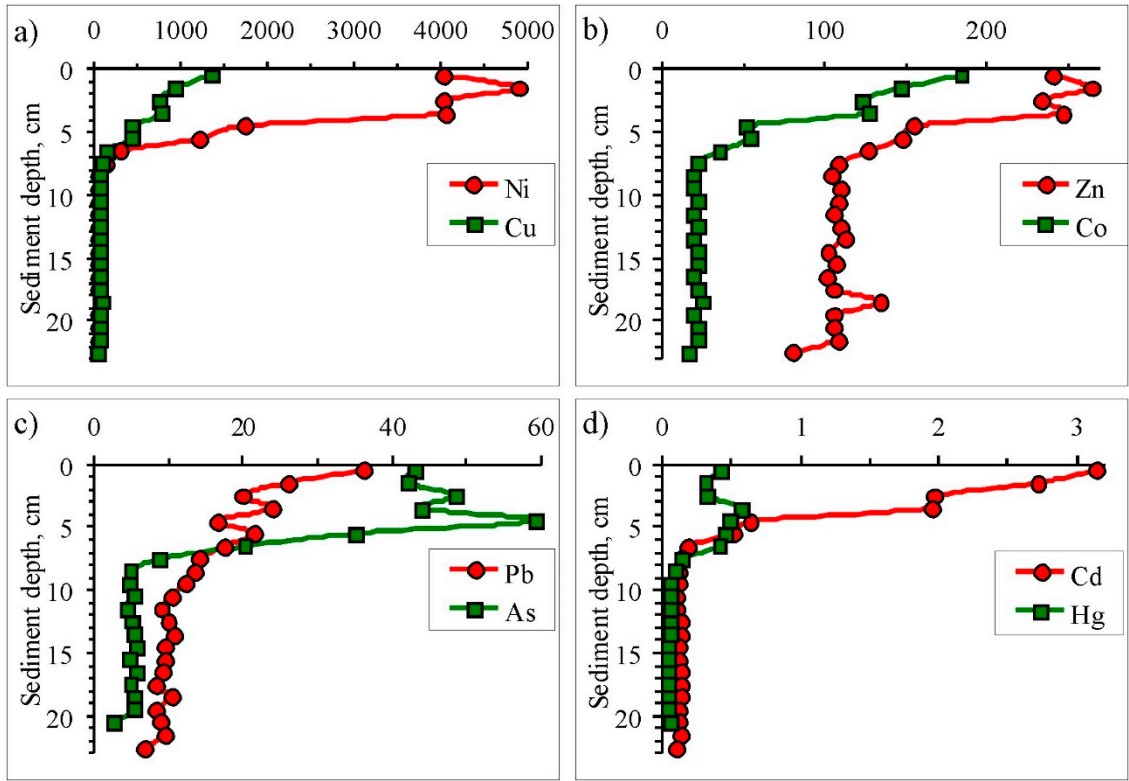

**Figure 5.** The vertical distribution of the content of elements (μg/g) in the sediments of Lake Kuetsjarvi (**a**: Ni and Cu, **b**: Zn and Co, **c**: Pb and As, **d**: Cd and Hg).

The values of the contamination factor $C_f$ of Ni, Cu, Cd, As, and Hg range from 8.5 to 125.7 (Table 2), i.e., are considered high pollution according to the classification proposed by L. Håkonson [20]. Ni, Cu, and Cd have the highest $C_f$ values. The contamination degree value $C_d$ (sum of the $C_f$ values of all the eight heavy metals) of the lake (236.2) is considered high.

**Table 2.** Concentrations of heavy metals (μg/g, dry weight) in the sediments of the Bely Kamen station (depth 32 m) from Lake Kuetsjarvi, pollution factor ($C_f$) and pollution degree ($C_d$).

| Layers of Sediments, cm | Ni | Cu | Zn | Co | Cd | Pb | As | Hg | $C_d$ |
|---|---|---|---|---|---|---|---|---|---|
| 0–1 | 4032 | 1343 | 240 | 184.1 | 3.14 | 36.1 | 43.1 | 0.417 | |
| 22–23 | 32 | 40 | 80 | 15.9 | 0.10 | 6.6 | 2.62 | 0.049 | |
| $C_f$ | 125.7 | 33.5 | 3.0 | 11.6 | 32.1 | 5.5 | 16.4 | 8.5 | 236.2 |

Despite the recent decrease in the anthropogenic load, Lake Kuetsjarvi has been one of the most polluted water bodies in the Pasvik River catchment in the past 30 years [1,46,47].

Analysis of mineral inclusions in the top (0–1 cm) layers of the sediments in Lake Kuetsjarvi revealed the presence in the sediments of anthropogenic particles sized 5 to 80 microns and having a spherical or irregular (often blob-like) shape (Figure 6). In their chemical composition, Fe, Ni, Cu, S, and O prevail, which indicates that their origins are associated with the processing of primary copper-nickel ore (crushing, firing, and smelting) [48]. Slag particles are also found where Si, Mg, and Ca are mixed with Fe oxide. The content of the lake's main pollutants Ni and Cu in these particles can reach 65% and 30%, respectively. Similar mineral anthropogenic formations were previously found in the sediments in Lake Nudjavr located near the city of Monchegorsk, where another copper–nickel smelter operates [49].

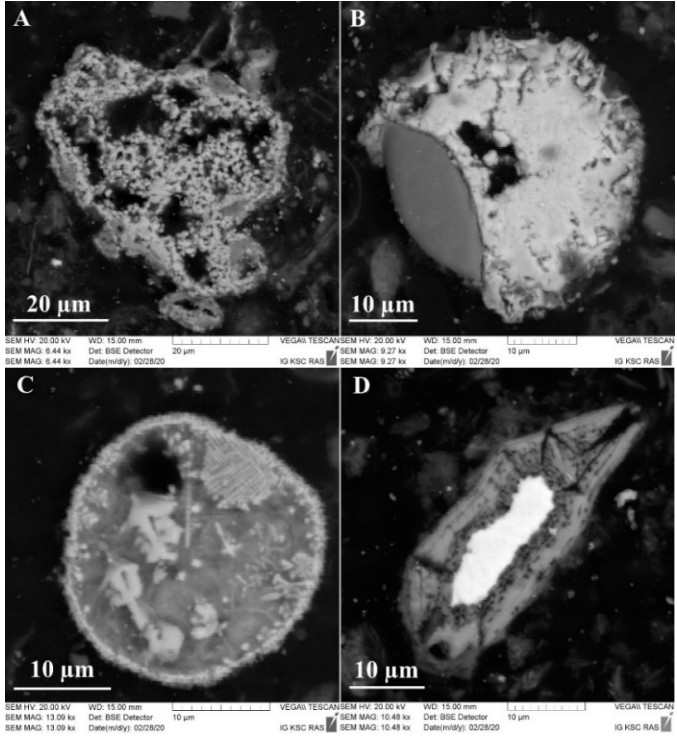

**Figure 6.** Images of technogenic microparticles in the sediments of Lake Kuetsjarvi, 2012 (**A**: Fe oxide with Ni, **B**: Fe sulfide with Ni, **C**: Fe oxide, **D**: Fe sulfide with Cu).

Plankton communities. Three large periods of the planktonic communities' development can be identified in Lake Kuetsjärvi, characterized by the composition of dominant taxa and quantitative indicators: 1st from 1994 to 1998; 2nd from 2007 to 2011; 3rd from 2012 to 2015. Average phytoplankton biomass and chlorophyll a values for the entire study period matched the β-mesotrophic trophic status Table 3). The long-term dynamics of plankton biomass and the ratio of the number of large taxonomic groups in the communities in different periods are shown in Figure 7a,b.

**Table 3.** Selected dynamics of the phytoplankton and zooplankton indicators of Kuetsjarvi Lake in different research periods (B—biomass (median, min-max), g/m$^3$; B3—biomass of predatory zooplankton, B2—filter feeding zooplankton; N—total abundance, thousand ind./m$^3$, Chl «a»—chlorophyll a content, mg/m$^3$ (median, min-max); H'(N)—Shannon biodiversity index by abundance, bits/ind; w = B/N—average individual zooplankton mass of the community, mg, S—saprobity index; K—water quality class; T—trophic status of the Lake).

| Indicators | Research Periods | | |
|---|---|---|---|
| | 1996–1998 | 2007–2011 | 2012–2015 |
| **Phytoplankton** | | | |
| Dominant taxa | *Melosira varians* <br> *Pandorina morum* <br> *Asterionella formosa* <br> *Diatoma tenuis* <br> *Dinobryon sociale* <br> *Eudorina* sp. | *Asterionella formosa* <br> *Fragilaria tenera var. nanana* <br> *Staurosira construens* <br> *Diatoma tenuis* <br> *Microcystis pulverea f. delicatissima* <br> *Dinobryon bavaricum* | *Asterionella formosa* <br> *Pseudosphaerocystis lacustris* <br> *Mucidosphaerium pulchellum* <br> *Fragilaria capucina* <br> *Pseudanabaena* sp. |
| B, g/m$^3$ | 1.06 <br> (1.06–2.05) | 1.95 <br> (1.33–2.65) | 2.30 <br> (1.23–10.68) |
| Chl «*a*», mg/m$^3$ | 4.23 <br> (2.71–5.95) | 4.51 <br> (0.35–4.04) | 5.06 <br> (0.46–28.34) |
| S | 1.92 <br> (1.02–1.44) | 1.27 <br> (1.17–1.52) | 1.42 <br> (1.30–1.88) |
| K | III | II | II |
| T | | β-mesotrophic | |

**Table 3.** *Cont.*

| Indicators | Research Periods | | |
| --- | --- | --- | --- |
| | 1996–1998 | 2007–2011 | 2012–2015 |
| **Zooplankton** | | | |
| Dominant taxa | *Kellicottia longispina, Keratella cochlearis, Keratella quadrata, Notholca* sp., *Polyarthra* sp., *Bosmina obtusirostris, Daphnia cristata* | *Kellicottia longispina, Keratella cochlearis, Polyarthra* sp., *Bosmina obtusirostris* | *Keratella cochlearis, Notholca* sp., *Polyarthra* sp. |
| N, thousand. Ind./m$^3$ | 44.7–62.0 | 80.0–147.0 | 272.1–1254.3 |
| B, g/m$^3$ | 0.2 (0.18–0.25) | 0.15 (0.02–0.68) | 2.3 (1.3–3.5) |
| H′(N) bit/ind. | 2.0–2.6 | 1.7–2.0 | 1.1–2.0 |
| B$_3$/B$_2$ | 0.1–0.5 | 0.01–0.5 | 0.01–0.9 |
| W = B/N, mg | 0.003–0.006 | 0.001–0.005 | 0.001–0.005 |
| T | α-oligotrophic | transitional from α- to β-oligotrophic | transitional from α- to β-mesotrophic |

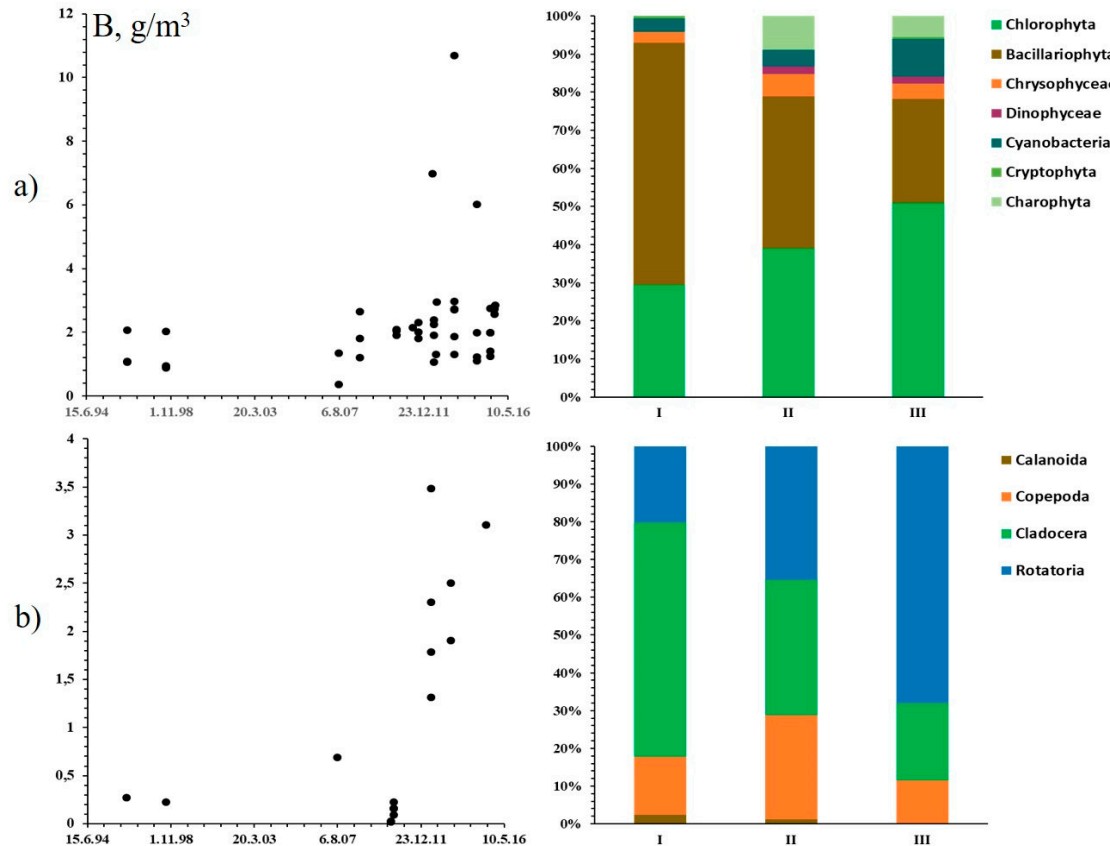

**Figure 7.** The long-term dynamics of plankton biomass and the ratio of the number of large taxonomic categories in the composition of communities in different periods of research (I—1996–1998; II—2007–2011; III—2012–2015). (**a**) phytoplankton; (**b**) zooplankton.

Sharov [35,36] showed that during the 1st period, diatoms dominated in the southern part and green algae (*Pandorina morum* (Müll.) Bory, 1827) dominated in the northern part of the lake and that golden algae (Dinobryon sociale (Ehrb.) Ehrb., 1834) were numerous. In the 2nd period, changes in the structure of dominance occurred—the share of cyanoprokaryotes ((Microcystis pulverea f. delicatissima W. and GSWest) Elenk., 1938) and Fragilariaceae species increased in the context of an increase in average biomass and chlorophyll a. Similarly to the 1st period, the share of green algae

(up to 39%) was high in the northern part of the lake. Further changes in phytoplankton communities (3rd period) were associated with an increase in the population of cyanoprokaryotes (*Pseudanabaena* sp.), especially towards the end of the growing season, and green algae (up to 51%), whose proportion increased not only in the northern, but also in the southern part of the lake (Figure 7a). The share of golden algae and diatoms shrank, along with an increase in biomass and chlorophyll a. The trend toward an increase in the phytoplankton biomass in Lake Kuetsjarvi between 1994 and 2015 indicates the intensification of primary production processes in the lake: the average biomass value doubled (Table 3). The Mann–Whitney U-test showed the statistical 249 confidence in phytoplankton biomass between 1st and 3rd periods. Differences between 1st and 2nd, 250 as well as 2nd and 3rd are not significant. At the same time, the maximum values of seasonal phytoplankton biomass also increased up to 10.68 g/m$^3$ (Figure 7a).

Phytoplankton dynamics in the hydrobiological summer season (in 2012–2013) were characterized by a high level of biomass (up to 7 g/m$^3$) beginning in June, which is characteristic of mesotrophic water bodies, with green algae representing the largest share (over 40%) in the composition of the communities. In July, biomass remained at the same level, the share of green algae increased (over 53%), with Volvocaceae and Chlamydomonas being the most abundant; the share of diatoms decreased, while dinophytic and golden algae almost disappeared. In July, the share of cyanoprokaryotes *Pseudanabaena* sp. also grew. By August, the algal biomass decreased, mainly due to the slowdown of the green algae vegetation processes (down to 1.4 g/m$^3$); golden algae ones completely disappeared from the communities, the share of cyanoprokaryotes shrank; diatoms were as abundant as before.

The combined effect of climate change and environmental pollution is complex, causing deep restructuring in the Arctic freshwater ecosystems. The most significant changes are observed in phytoplankton communities: over the past 20 years, there has been a change in the dominant taxa, the share of green algae and cyanoprokaryotes has increased along with a decrease in the share of diatoms and golden algae characteristic of the Arctic, while the average phytoplankton biomass was several times higher than the reference values. The values of the saprobity index indicate a change in the water quality class from 3 to 2. Probably, this does not mean intensified self-purification of the water reservoir, but, on the contrary, indicates the growth of the pollution load, when exposed to which species resistant to pollution start to proliferate. This is supported by the long-term trends toward an increase in the level of heavy metals in water. A strong factor of the observed change is also the warming of the Arctic climate, which can amplify the effects of eutrophication in water bodies [7]. The decrease in the saprobity index illustrates changes in the water hydrochemistry, in particular, the ratio of pollution and trophic load on the lake against the background of climate warming. The summer dynamics of phytoplankton in Lake Kuetsjarvi are untypical of subarctic lakes, due to the intensive growth of green algae, which begin to actively vegetate already in June.

In the 1st period, the lake's zooplankton community is characterized by low abundance and biomass—44.7 to 62.0 thousand individuals per m$^3$ and 0.2 g/m$^3$, respectively (Table 3). Rotifers *Kellicottia longispina* Kellicott, 1879, *Keratella cochlearis* Gosse, 1851, *Keratella quadrata* Müller, 1786, *Notholca* sp., *Polyarthra* sp., and cladocerans *Bosmina obtusirostris* Sars, 1862, and *Daphnia cristata* Sars, 1862, prevailed. The ratio of the main taxonomic groups Rotifera:Cladocera:Copepoda reflects the prevalence of rotifers in terms of abundance (74.7%–87.6%) and Cladocera in terms of biomass (58.4%–79.9%) (Table 3). Herbivores prevailed over predatory species (B$_3$/B$_2$ < 1). The Shannon species diversity index (based on abundance H(N)) varied within 2.0–2.6 bit/individual; the average individual biomass in the zooplankton community (w = B/N) was 0.003–0.006 mg. The lake's trophic status was considered very low—α-oligotrophic (Table 3).

The study of the zooplankton community indicators in the 2nd period revealed an increase in the total number of organisms by a factor of 2 (from 80.0 to 147.0 thousand individuals per m$^3$) (Table 3). The total biomass was comparable with that of the previous period and amounted to 0.1–0.2 g/m$^3$, except for the northern part of the lake (Gulf Stream Station, 0.7 g/m$^3$), where the valuable as prey species large cladocerans and copepods were found (47.2% and 43.2% of the total biomass, respectively).

Herbivore species still prevailed over predators ($B_3/B_2 < 1$); rotifers *Kellicottia longispina*, *Keratella cochlearis*, *Polyarthra* sp., and Cladocera *Bosmina obtusirostris* dominated. The Shannon index fell to 1.7–2.0 bit/individual; the average individual biomass in the zooplankton community (w = B/N) was also low at 0.001–0.005 mg. The lake's trophicity type is transitional from very low α-oligotrophic to low β-oligotrophic (Table 3).

Period 3 is characterized by abnormally high quantitative characteristics of the zooplankton. Total abundance increased to 722.8–1254.3 thousand individuals/m$^3$, biomass to 1.3–3.5 g/m$^3$ (Table 3). Rotifers prevailed at >90% of the total abundance and 75% of the total biomass of the organisms, still dominated by *Keratella cochlearis*, *Notholca* sp., and *Polyarthra* sp. Few individuals of the Cladocera species *Bosmina obtusirostris*, one of the core structure-forming species in the earlier study periods, were observed in the samples. An analysis of the ratio of the main taxonomic groups to the total biomass in a long-term series of observations reflects the prevalence of rotifers in the 3rd study period (Figure 7b). The Shannon species diversity index is low (1.1–2.0 bit/individual), which indicates an increase in the dominance of individual Rotifera species. Herbivores prevailed over predatory species ($B_3/B_2 < 1$). The average individual biomass in the zooplankton community was comparable to that in the two previous periods and ranged within 0.001–0.005 mg, which also reflects the large-scale development of small-sized species with simple life cycles and high reproduction rates (rotifers).

The modern zooplankton community of Lake Kuetsjarvi shows a decrease in the abundance of typical palearctic species and an increase in the abundance of *r*-strategist–eurybiontic species with high ecological valence, simple life cycles, and a high reproduction rate (rotifers). The development of monocultures composed of pollution-resistant species is observed. Valuable prey Cladocera species ("fine" filtrators belonging to the *Bosmina* and *Daphnia* genera) dominated only in the first (1996–1998) and second (2007–2010) study periods. Few pollution-sensitive active "coarse" filtrators (copepods *Eudiaptomus gracilis*, Sars, 1863, *Eudiaptomus graciloides* Lilljeborg, 1888) who play an important role in the process of self-purification of water, and *Cyclopiformes* species were found in the samples. In the context of fluctuating levels of multifactorial anthropogenic load and climate change, a steady trend is observed toward a reduced species diversity of the zooplankton communities in subarctic water bodies, simpler structures thereof, change in dominant species, and the development of monocultures composed of pollution-resistant organisms.

The trophic status of the lake, according to the trophic scale proposed by Kitaev [34], has grown in a long-term observation series from very low α-oligotrophic to medium β-mesotrophic. This indicates an unfavorable environmental situation in the lake and a reduced biofiltration activity of the zooplankton as a whole, especially in the 3rd period, starting in 2012, which is the result of the combined effects of natural and anthropogenic factors against the backdrop of climate warming.

Benthic macroinvertebrates. The Benthic communities of Lake Kuetsjarvi in 1990–1992 were characterized by high species diversity—56 species and forms of invertebrates were found in the water reservoir. The most diverse were the Chironomidae, which accounted for the bulk of quantitative indicators (>70% of the species composition and 80% of the total zoobenthos abundance). In the littoral zone, Chironomidae *Cricotopus*, *Stictochironomus*, *Orthocladius*, *Tanytarsini*, caddisflies *Limnephilidae* and *Polycentropodidae*, mollusks *Pisidium sp.*, and water bugs were common. In the profundal zone (at depths of 14–15 m), Chironomidae *Chironomus sp.* and *Sergentia sp.*, oligochaete *Tubifex tubifex* Müller, 1774, dominated and a small number of mollusks *Pisidium sp.* was found. Average abundance and biomass values of deepwater zoobenthos in Lake Kuetsjarvi (at depths of 15–20 m) in 1990–1992 were 2267 (min–max 172–4633) individuals/m$^2$ and 12.1 (min–max 0.5–21.1) g/m$^2$ [10,50] (Table 4).

**Table 4.** Some indicators of benthic communities of Lake Kuetsjärvi (N—abundance, ind./m$^2$; B—biomass, g/m$^2$; T—trophic status of water) at different research period.

| Indicators | Research Period | |
| --- | --- | --- |
| | 1990–1992 * | 2009–2013 |
| Dominantspecies | *Cricotopus* ** *Stictochironomus*, *Orthocladius*, *Tanytarsini*, *Pisidium* sp. *Limnephilidae*, *Polycentropodidae* *Chironomus* sp., *Sergentia* sp., *Tubifex tubifex* | *Cricotopus silvestris* gr. *Procladius choreus* gr. *Tubifex tubifex* *Sergentia coracina* *Chironomus* sp. *Prodiamesa olivacea* |
| N, ind./m$^2$ | - *** 2267 | 1680 506 |
| B, g/m$^2$ | - 12.1 | 7.5 2.1 |
| Total number of taxa | 56 | 28 |
| T | eutrophic | oligotrophic |

Note: * 1990–1992 гг.—Lukin et al., 2003; 2009–2013 гг.—own data; ** in the numerator—littoral zone, in the denominator—profundal zone; ***—no data.

According to the findings of the studies conducted in 2009–2013, the diversity of benthic fauna in Lake Kuetsjarvi had not changed; 28 species and supraspecific taxa were identified. The groups most sensitive to contamination—mayflies, stoneflies, leeches, and crustaceans—were not found in the samples, similar to the previous study period.

Benthic fauna abundance in the coastal zone averaged 1680 individuals/m$^2$, biomass 7.5 g/m$^2$. Chironomidae *Cricotopus silvestris* gr., and *Procladius choreus* gr. formed the basis of the littoral communities of zoobenthos, caddisflies, and hemipterans were subdominant. With increasing depth, the diversity of fauna and quantitative indicators of zoobenthos decreased. The basis of the bottom fauna in the deepwater areas of Lake Kuetsjarvi was formed by the Chironomidae *Sergentia coracina* Zett., 1850, *Chironomus* sp., *Prodiamesa olivacea* Meigen, 1818 (30%–45% of the total abundance and 50%–70% of the total zoobenthos biomass) and the oligochaete *T. tubifex* (53% and 27%, respectively). The abundance of zoobenthos in the profundal zone averaged 506 (min–max 69–1660) individuals/m$^2$, biomass 2.1 (min–max 0.3–8.3) g/m$^2$ (Table 4).

Ichthyofauna. Given the selected periods when analyzing the state of planktonic communities, the ichthyological data collected on Lake Kuetsjarvi over the past 25 years has been conditionally subdivided into three study periods: 1990–1998 (period 1), 2004–2009 (period 2), and 2012–2015 (period 3).

The lake's ichthyofauna is composed of eight indigenous species belonging to eight families of fish: trout *Salmo trutta* Linnaeus, 1758, whitefish *Coregonus lavaretus* (Linnaeus, 1758), grayling *Thymallus thymallus* (Linnaeus, 1758), pike *Esox lucius* Linnaeus, 1758, perch *Perca fluviatilis* Linnaeus, 1758, burbot *Lota lota* (Linnaeus, 1758), minnow *Phoxinus phoxinus* (Linnaeus, 1758), and nine-spined stickleback *Pungitius pungitius* (Linnaeus, 1758). Among the invasive species, the European vendace *C. albula* (Linnaeus, 1758) deserves to be mentioned, which was first introduced in the 1960s in the Finnish Lake Inari and subsequently (from the early 1990s) has spread throughout the catchment area of the Pasvik River [12,51]. Whitefish dominated catches in all study periods (Table 5). Other numerous species in the lake were perch, pike, and invasive vendace. Starting from the 2nd study period, there has been a trend toward a reduced down to 77% share of whitefish in catches (Table 5). The share of pike in catches during the same period decreased more twice, while the share of perch and vendace increased. Given the above, the dynamics of the biological indicators of whitefish, vendace, pike, and perch is given below.

**Table 5.** Composition of fish catches,% in Lake Kuetsjarvi, 1990–2015.

| Fish Species | Research Period | | |
|---|---|---|---|
| | 1990–1998 | 2004–2009 | 2013–2015 |
| Trout | 0.5 | 0.3 | 0.8 |
| Whitefish | 85.7 | 80.5 | 77.4 |
| Vendace | 0.0 | 3.9 | 7.3 |
| Grayling | 0.1 | 0.1 | 0.0 |
| Pike | 6.2 | 1.3 | 2.5 |
| Perch | 6.9 | 13.3 | 10.8 |
| Burbot | 0.6 | 0.6 | 1.2 |

During all study periods, whitefish were represented by two morphs: sparsely rakered (15–35 gill rakers, average 23.5 ± 0.14) and densely rakered (27–44 gill rakers, average 33.2 ± 0.08) (Figure 8a–c). This is typical for the whitefish population of the Inari-Pasvik system as a whole [12,51]. The actual ratio of the two whitefish morphs in Lake Kuetsjarvi can be judged from samples from the study periods 1 and 2, where the gill apparatus was examined in 99% of individuals. In the 2nd period, the ratio of sparsely rakered and densely rakered whitefish in catches averaged 1 (263 individuals):1 (308 individuals), while in the 3rd period, densely rakered whitefish prevailed in catches at a ratio of 2 (400 individuals):1 (213 individuals).

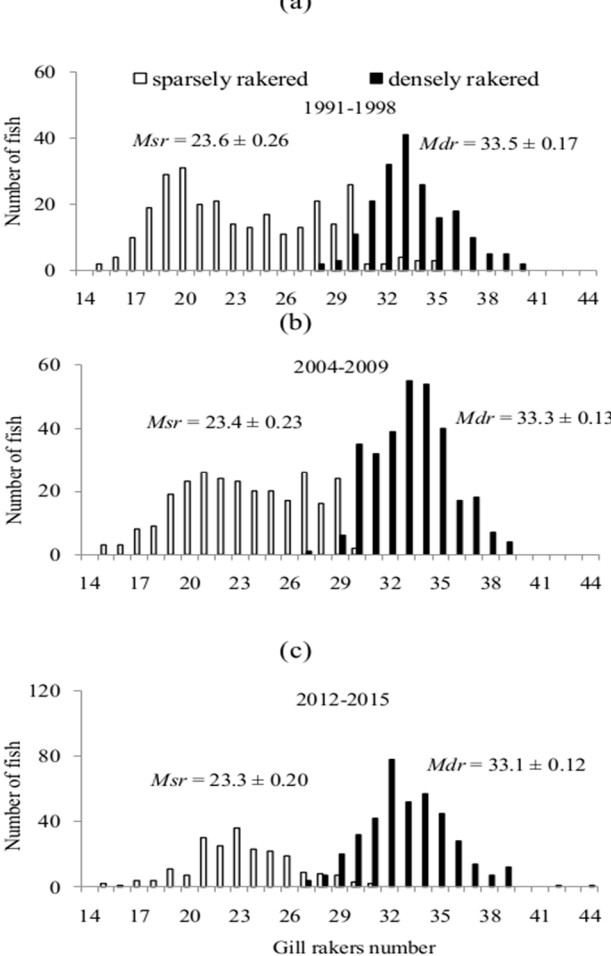

**Figure 8.** The distribution of whitefish by the number of gill rakes on the first branchial arch in Lake Kuetsjarvi in: 1991–1998 (**a**), 2004–2009 (**b**), 2012–2015 (**c**). M ± m is the mean value and its error, sr—sparsely rakered whitefish, dr—densely rakered whitefish.

The average and extreme linear and weight characteristics for samples of two forms of whitefish, vendace, perch, and pike for different periods of the study are presented in Table 6. The average body sizes calculated over the entire sample for the period are determined, first of all, by the age structure of the sample and the dynamics of growth rates, due to a combination of environmental factors. This is seen in Figure 9a–f, where the distribution of length and weight varies depending on the year, and therefore the average estimates will shift depending on the number of dominant age groups and growth rates in the samples. High levels of heavy metals in water affect the mineral metabolism in fish, causing various deformations of bone tissue, including irregularities in age-recording structures. Often this makes it impossible to determine the age of all studied specimens and/or increases the error of these definitions, amplified by subjective circumstances. Therefore, accurately measured linear-weight indicators seem to us more reliable. Age of vendace, perch, and pike in Lake Kuetsjarvi is defined partially. The greatest age determinations were obtained for whitefish. As a result, we were able to compare linearly and weight characteristics of fish at different ages only in whitefish for 1 and 3 periods of the study (Table 7). As can be seen from the table, the sparsely rakered and densely rakered whitefish in different periods of the study had a close age composition, represented by a limited number of age groups. The maximum age of whitefish was 10+, but fish older than 8+ were represented by single specimens. In sparsely rakered whitefish, in both periods, 2 + −4 + ages fish predominated, in the densely rakered whitefish, in the 1st period −1 + −3 + ages fish, and in 3rd period—older fish. At the same time, both forms of whitefish grew significantly faster in the 1st period than in the 3rd (Table 7). Thus, we are currently observing a decrease in the linear and weight growth rates of whitefish in Lake Kuetsjarvi.

**Table 6.** Average size (*FL*), mm and weight, g characteristics of fish as a whole for the sampling and sexually mature individuals in Lake Kuetsjarvi, 1990–2015.

| Fish Species | Length as a Whole for the Sampling, mm | Weight as a Whole for the Sampling, g | Length of Sexually Mature Fish, mm | Weight of Sexually Mature Fish, g |
|---|---|---|---|---|
| **1990–1998** | | | | |
| Sparsely rakered whitefish | 195 ± 3.7 \ 88–465 (318) | 117 ± 9.8 \ 4–1300 (318) | 206 ± 7.7 \ 90–465(42) | 175 ± 22.2 \ 4–1300 (42) |
| Densely rakered whitefish | 139 ± 3.3 \ 85–300 (211) | 34 ± 3.4 \ 4–278 (211) | 146 ± 5.2 \ 93–280 (36) | 46 ± 5.9 \ 5–261 (36) |
| Vendace | - | - | - | - |
| Pike | 384 ± 7.1 \ 255–540 (81) | 487 ± 26.9 \ 110–1205 (81) | 427 ± 15.6 \ 310–540 (18) | 620 ± 63.1 \ 207–1205 (18) |
| Perch | - | 98 ± 9.2 \ 6–385 (90) | - | - |
| **2004–2008** | | | | |
| Sparsely rakered whitefish | 160 ± 1.8 \ 83–265 (421) | 49 ± 1.8 \ 6–254 (421) | 161 ± 2.1 \ 87–260 (228) | 48 ± 2.2 \ 6–210 (228) |
| Densely rakered whitefish | 134 ± 2.4 \ 75–293 (359) | 33 ± 2.1 \ 2–340 (359) | 133 ± 3.1 \ 87–293 (218) | 35 ± 2.9 \ 4–340 (218) |
| Vendace | 110 ± 1.6 \ 100–130 (38) | 11 ± 0.5 \ 6–19 (38) | 109 ± 1.7 \ 100–130 (34) | 11 ± 0.5 \ 6–19 (34) |
| Pike | 326 ± 18.3 \ 255–499 (13) | 373 ± 71.5 \ 164–1079 (13) | 499 (1) | 890 (1) |
| Perch | 186 ± 4.2 \ 110–295 (129) | 122 ± 10.7 \ 17–604 (129) | 252 ± 5.2 \ 128–295 (33) | 296 ± 17.9 \ 28–604 (33) |
| **2012–2015** | | | | |
| Sparsely rakered whitefish | 185 ± 3.6 \ 77–407 (233) | 94 ± 8.2 \ 4–981 (233) | 188 ± 2.62 \ 116–407 (89) | 95 ± 13.33 \ 14–843 (89) |
| Densely rakered whitefish | 160 ± 2.58 \ 75–312 (416) | 58 ± 3.0 \ 4–365 (416) | 160 ± 5.0 \ 80–312 (144) | 64 ± 6.53 \ 4–365 (144) |
| Vendace | 104 ± 1.1 \ 87–141 (62) | 9 ± 0.4 \ 5–24 (62) | 104 ± 1.0 \ 96–123 (37) | 9 ± 0.5 \ 6–19 (37) |
| Pike | 408 ± 15.7 \ 260–547 (21) | 637 ± 78.5 \ 150–1385 (21) | 449 ± 24.3 \ 400–516 (4) | 788 ± 158.5 \ 471–1220 (4) |
| Perch | 155 ± 4.9 \ 45–290 (91) | 77 ± 10.7 \ 1–792 (91) | 169 ± 7.8 \ 119–290 (28) | 87 ± 17.2 \ 23–440 (28) |

Note: here and in Table 7, above the line—the average value and its error, below the line—the limits of variation of the indicator, in brackets—the number of fish, individuals.

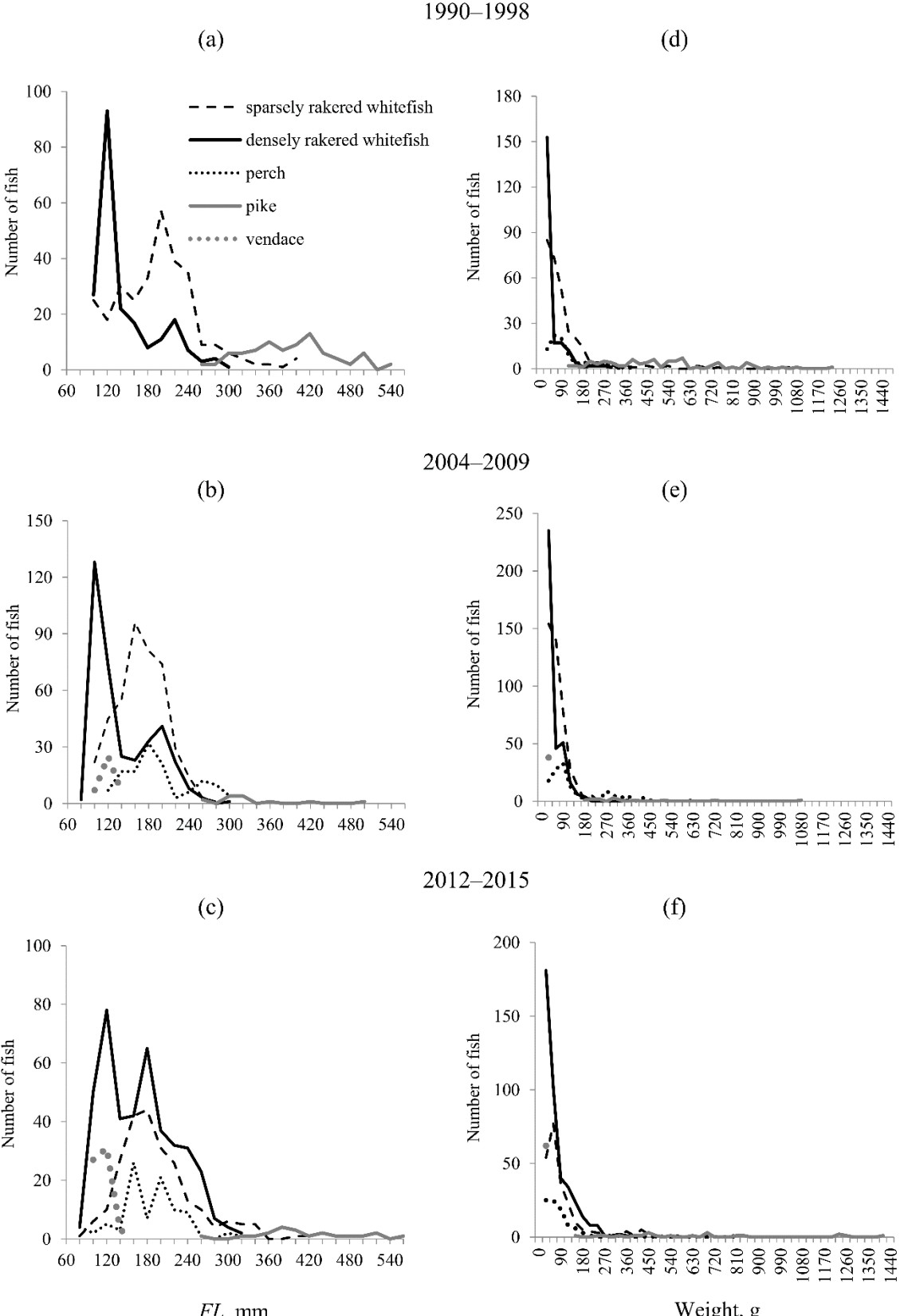

**Figure 9.** The frequency distribution of whitefish, vendace, pike, and perch along the length (*FL*), mm (**a**–**c**) and weight, g (**d**–**f**) in the Lake Kuetsjarvi, 1990–2015.

**Table 7.** Comparison of length (*FL*), mm, and mass (*W*), g of various forms of whitefish in 1991–1998 and 2012–2015.

| Fish Species, Study Period | Age | | | | | | | | | |
|---|---|---|---|---|---|---|---|---|---|---|
| | 1+ | 2+ | 3+ | 4+ | 5+ | 6+ | 7+ | 8+ | 9+ | 10+ |
| | *FL* | | | | | | | | | |
| Sparsely rakered whitefish, 1990–1998 | 136 ± 18.9 103–220 (6) | 171 ± 7.0 97–235 (23) | 195 ± 5.5 124–250 (35) | 231 ± 6.4 173–325 (30) | 259 ± 9.6 206–330 (16) | 321 ± 7.3 289–343 (7) | 388 ± 9.5 360–400 (4) | 400 (1) | 285 (1) | 465 (1) |
| Sparsely rakered whitefish, 2012–2015 | 103 ± 7.0 77–137 (7) | 148 ± 4.8 84–215 (43) | 156 ± 3.8 112–237 (52) | 188 ± 4.8 124–270 (53) | 231 ± 9.2 160–334 (35) | 220 ± 12.1 162–407 (22) | 221 ± 14.8 190–292 (6) | 421 (1) | 214 ± 11.5 202–225 (2) | 362 (1) |
| T | 1.77 | **2.73** | **5.93** | **5.37** | 1.81 | **4.59** | **8.33** | - | - | - |
| Densely rakered whitefish, 1990–1998 | 117 ± 3.9 102–158 (16) | 159 ± 7.3 110–220 (19) | 207 ± 8.0 165–270 (13) | 266 ± 5.2 255–275 (4) | 202 ± 8.0 194–210 (2) | - | 300 (1) | - | - | - |
| Densely rakered whitefish, 2012–2015 | 97 ± 5.5 91–102 (2) | 126 ± 2.6 80–253 (139) | 156 ± 3.4 97–247 (88) | 184 ± 4.9 107–247 (72) | 225 ± 3.9 131–310 (58) | 261 ± 9.2 214–312 (12) | 259 ± 16.0 243–275 (2) | - | 295 (1) | - |
| T | 1.75 | **4.38** | **5.45** | **3.95** | 1.09 | - | - | - | - | - |
| | *W* | | | | | | | | | |
| Sparsely rakered whitefish, 1990–1998 | 33 ± 16.5 8–113 (6) | 58 ± 7.4 7–142 (23) | 89 ± 8.3 17–189 (35) | 153 ± 18.5 24–465 (30) | 233 ± 31.5 106–475 (16) | 468 ± 33.6 284–536 (7) | 853 ± 71.8 658–1000 (4) | 1037 (1) | 277 (1) | 1300 (1) |
| Sparsely rakered whitefish, 2012–2015 | 10 ± 2.4 4–24 (7) | 36 ± 4.0 5–115 (43) | 41 ± 3.7 13–135 (52) | 81 ± 8.1 16–300 (53) | 173 ± 23.9 40–488 (35) | 274 ± 52.3 86–1295 (22) | 133± 34.6 80–302 (6) | 981 (1) | 118 ± 19.5 98–137 (2) | 440 (1) |
| T | 1.47 | **2.92** | **5.84** | **4.12** | 1.47 | 2.03 | **10.09** | - | - | - |
| Densely rakered whitefish, 1990–1998 | 14 ± 2.1 6–36 (16) | 43 ± 6.7 8–118 (16) | 99 ± 13.9 38–226 (13) | 224 ± 20.1 182–261 (4) | 69 ± 31.1 47–91 (2) | - | 278 (1) | - | - | - |
| Densely rakered whitefish, 2012–2015 | 8 ± 0.5 7–8 (2) | 22.3 ± 1.8 4–176 (139) | 43 ± 3.0 7–189 (88) | 74 ± 5.2 11–187 (72) | 134 ± 7.5 24–365 (58) | 213 ± 23.9 117–349 (12) | 212 ± 52.0 160–264 (2) | - | 290 (1) | - |
| T | 1.11 | **3.71** | **5.91** | **6.67** | 1.60 | | | | - | |

Note: T - Student's coefficient, significant differences are highlighted in bold.

In the densely rakered whitefish distribution by body length and weight in all study periods, two peaks are clearly distinguished (Figure 9a–f), which makes it possible to conventionally distinguish small and large whitefish individuals. In the sparsely rakered whitefish, the distribution pattern by body size and weight in period 1 is more complex with several pronounced peaks, while in the other periods only one peak can be seen (Figure 9a–f). A detailed study of the linear growth rates of whitefish in Lake Kuetsjarvi in 2015 identified additional groups in the sparsely rakered and densely rakered whitefish: small and large [15]. The selected groups differ significantly in the rates of linear growth throughout life.

As with whitefish, several peaks can be distinguished in the frequency distribution of perch along the length and mass (Figure 9b–f). Moreover, in periods 1 and 3, the distribution peaks of individuals have a different arrangement.

The average and extreme linear and weight characteristics of mature fish for samples are presented in Table 6. Individuals of both whitefish morphs and vendace in Lake Kuetsjarvi in all three study periods started to mature at a similar body length and weight: 80–116 mm and 4–14 g, respectively. Lake Kuetsjarvi perch starts to mature at a body length and weight of 119–128 mm and 23–28 g, respectively, pike at a body length of 310–400 mm and a body weight of 207–471 g (Table 6).

Changes in the structure of the fish portion of the lake's community are expressed in a decrease in the proportion of oligotoxobic whitefish [12,52,53] and an increase in the proportion of eurybiontic perch and invasive vendace. Although vendace was first registered in Lake Kuetsjarvi in 1995 [51], its numbers in the following ten years did not grow as fast as it was observed in the upper reaches of the Pasvik River system (Ruskebukta) in 1991–1995. This is most likely due to the lack of suitable spawning sites for vendace in the lake, and its population is maintained by the fish migrating from the Pasvik River. Densely rakered whitefish is dominant in the pelagic zone of Lake Kuetsjarvi (54%) (based on 2015 catches), vendace accounts for 29%, perch for the remaining 17%.

Over the past 30 years, Lake Kuetsjarvi has been one of the most polluted water bodies in Murmansk Region, whitefish are in high abundance and a complex population structure, characteristic of the Inari-Pasvik system as a whole [54–63]: sparsely rakered and densely rakered morphs have developed and formed subgroups that differ in growth rates (smaller and larger fish individuals) [15]. Considering the above-described changes in the ratio of intraspecific whitefish groups in the different study periods in Lake Kuetsjarvi, at this stage, it may be reasonable to speak of both a single polymorphic whitefish population, composed of dissimilar individuals, cross-breeding between which is highly likely, and about a group of reproductively isolated populations [64].

Generally, the slowly growing sparsely rakered and densely rakered whitefish individuals in Lake Kuetsjarvi become sexually mature at the lowest body length and weight values for this species (see above), which is considered one of the forms of adaptation of whitefish to the extreme conditions in the lake [12].

According to the catch data from 2015, perch in Lake Kuetsjarvi equally populates both the pelagic (50%) and profundal (50%) ecological zones of the water reservoir. In perch, the development of two subspecific forms within the same population is also observed here, which differ in growth rates. Given the changes in the ratio of body sizes of two subspecies forms of perch at different periods of the study, it is likely that one of the forms passes to the other during ontogenesis [65].

In connection with the closure of the smelter at the shores of the lake, it seems rather interesting to continue monitoring the fish community to study the responses of various levels to the inevitable positive changes in the environment. A likely scenario for the ecosystem's response to a reduction in the heavy metal load may be increased lake eutrophication and perch dominance in the fish community.

## 4. Conclusions

Surface waters are among the most important natural resources of the Arctic. Their resource potential is controlled not only by quantitative indicators, but also by qualitative ones, including biological indicators of water quality, structural, and functional indicators of hydrobiont communities.

The concept of using bioindicators and biomarkers for assessing environmental risks has a special place, since these directly reflect the responses of biological systems on various levels to the effects of bioavailable forms of chemical compounds present in the environment, integrate the effects of multicomponent effects, their spatial and temporal dynamics [12,66]. Recently, more and more attention has been paid to the evolutionary aspects (evolutionary ecotoxicology), which take into account the possibility of local adaptation of biological systems to the prolonged exposure to environmental stresses [66–69]. Moreover, most of the research is devoted to studying the functional responses at the organismal or suborganismal levels (molecular, biochemical, cellular, physiological, and behavioral) to the effects of one or more pollutants. Significantly fewer studies exist of the responses of natural populations and/or communities in general to long-term anthropogenic impacts, because such studies require long-term observations, and their results are more difficult to interpret due to lower specificity and greater inertia of these indicators [12].

Long-term comprehensive studies of the ecosystem of Lake Kuetsjarvi have made it possible to identify the response of its components to the global and regional change in the environment and climate as a whole, resulting in increased water toxicity and eutrophication, reduction in the number of stenobiont species of aquatic organisms against the background of an increase in the number of eurybiontic and invasive species. Environmental protection measures adopted by the Pechenganickel Smelter in recent decades have led to a reduction in the acidification load on the natural environment, but the heavy metal load has increased. Increased levels of nutrients in combination with the pronounced climate change trends [7] accelerate the processes of eutrophication. Modern communities of Lake Kuetsjarvi are the result of a combination of long-term changes in the abiotic environment and biotic interactions. Heavy-metal pollution of Lake Kuetsjarvi, observed since the 1930s, has led to the formation of a community that is resistant to this type of impact and supports large populations of adapted species. The identified trends indicate a radical restructuring of the structural and functional organization of the ecosystem. Adaptations of communities to the dynamics of the anthropogenic load causing changes in the environmental conditions of species habitats include changes in the species composition, quantitative indicators, ratios between individual taxonomic groups, and population structures.

Unlike the organismal level, responses to medium-term environmental changes on the population and community level are less specific and characterized by stronger inertia. The same trend is observed with an increase in the trophic level of biological systems. The identified key directions of change in the structure of the fish community of Lake Kuetsjarvi are similar to the changes observed in large water bodies in the Russian Arctic under changing environmental conditions (change in climatic and hydrological conditions, eutrophication, and pollution), manifested in a weaker role of long-cycle stenobiont autumn-spawning salmon and whitefish species and an increase in the abundance of eurybotic spring-spawning species of low commercial importance. However, the pace of these structural changes is much lower than in plankton communities. Despite the distinctive natural conditions and anthropogenic pollution in the lake, the development of sympatric forms that differ in the ecological niches they occupy, morphology, and life cycle strategies, including the transition to a short-cycle survival strategy, allows whitefish to remain the dominant species and maintain high population numbers.

**Author Contributions:** Conceptualization, E.M.Z. and N.A.K.; data curation, E.M.Z.; investigation, V.A.D., D.B.D., S.A.V., O.I.V., Z.I.S., P.M.T., and A.A.C. All authors have read and agreed to the published version of the manuscript.

**Funding:** This research was funded by research project 0226-2019-0045 and partially supported by the RFBR grant 18-05-60125 Arctic and RSF grant 19-77-10007. Interpretation of hydrological data was funded as part of the research project 0226-2019-0045; interpretation of hydrochemical and sediment data was funded as part of the RSF grant 19-77-10007; interpretation of hydrobiological and ichthyological data was funded as part of the RFBR grant 18-05-60125 Arctic.

**Conflicts of Interest:** The authors declare no conflict of interest. The funders had no role in the design of the study; in the collection, analyses, or interpretation of data; in the writing of the manuscript, or in the decision to publish the results.

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
