# Peer review of "Long-Term Environmental Monitoring in an Arctic Lake Polluted by Metals under Climate Change"

_environments, doi:10.3390/environments7050034_

Round 1

Reviewer 1 Report

You may consider studying the microbenthos, such as, benthic foraminifera as they are very sensitive and very good proxies for environmental changes.

Author Response

Dear Reviewer,
Thank you very much for the work done on our article.
Your opinion was important to us.

We answered your comment: "You may consider studying the microbenthos, such as, benthic foraminifera as they are very sensitive and very good proxies for environmental changes". 

Foraminifera is an interesting object for assessing environmental changes, but these are marine organisms. The laboratory is currently studying freshwater bodies of water. The object of research in this article is a freshwater lake. 

Best regards!

Elena Zubova

Reviewer 2 Report

We recommend to use acronyms for long names of institutes:

"90 Institute of Industrial Ecology of the North, Kola Research Center of the Russian Academy of Sciences" 

like Finnish Environment Institute (SYKE).

On methodology need to name a method or standard for example on sediment sample like:  

"91 Sediment core sampling, sample preparation, and chemical analysis methods are as described earlier [19]."

but earlier have mention just location of the station where the sediment sampling was done. in 99 line is described how the sediment where analysed by microscopy and spectrometry methods. 

125 Benthic samples were analyzed using recommended standard methods [25] - what method? how recommended?

it is better to have continuity of phrase not to have a general reference. 

Author Response

Dear Reviewer, thank you very much for the work you have done on our article!
Thank you for your valuable comments. We tried to correct our mistakes.

- "We recommend to use acronyms for long names of institutes".

We agree with the remark. Changes made to the text.

- ""On methodology need to name a method or standard for example on sediment sample like: "91 Sediment core sampling, sample preparation, and chemical analysis methods are as described earlier [19]."

Đ’ut earlier have mention just location of the station where the sediment sampling was done. in 99 line is described how the sediment where analysed by microscopy and spectrometry methods".

Big apologies, here is an inaccurate translation from Russian into English. We fixed the translation: «Sediment core sampling, sample preparation and chemical analysis methods have been described previously [19]".

- "Benthic samples were analyzed using recommended standard methods [25] - what method? how recommended?». 

We agree with the remark. Changes made to the text.

"Benthic samples were analyzed using standard methods recommended by the Russian State’s standards and Guidance on Methods [25, 22]. Animals were preserved in the field in 70% ethanol and sorted and identified later in the laboratory under a stereomicroscope".

Best regards,
Zubova Elena

Reviewer 3 Report

Originality/Novelty: Is the question original and well defined? Do the results provide an advance in current knowledge?

The study reports results of the long term environmental monitoring of the heavily polluted arctic lake Kuetsjarvi. The samples were collected between 1989 and 2016 and thus provide insights into the transformation of the lake ecosystem over the last 27 years. Zubova et al. have performed all-around studies of the biotic and abiotic components of the lake ecosystem and described the population structures and adaptations of lake communities to high concentrations of heavy metals in water and sediments. The study is of high interest and describes the environmental consequences of the metallurgical activities in the arctic.

Significance: Are the results interpreted appropriately? Are they significant? Are all conclusions justified and supported by the results? Are hypotheses and speculations carefully identified as such?

The study has clearly identified hypothesis and study methods are adequately selected for the purposes of the study. Collection of the samples and laboratory analyses are performed according to standardized procedures, in the internationally accredited analytical laboratory and thus the data collected over a long study period can be trusted. The section “Materials and methods” is well structured and contains a comprehensive overview of the methods used in the study. Where necessary the procedures are supplemented by references describing the methods to details. Significance of all reported data is supported by statistical analysis and the data have a sufficient level of statistical confidence to draw reliable biological conclusions. The detailed description of the statistical methods used in the study is however omitted. The later has to be corrected and a description of statistical procedures should be added to either each existing sub-section of the “materials and methods” or as a separate sub-section. The conclusions of the study are well justified and are all supported by measured environmental parameters and results of laboratory analyses. All speculations are referring to earlier published research papers and thus are compliant with prior knowledge.

Quality of Presentation: Is the article written in an appropriate way? Are the data and analyses presented appropriately? Are the highest standards for presentation of the results used?

The article is well structured and presented in accordance with the requirements of the journal. The figures are of good quality and clearly represent the findings of the study.

183

The signature to the x-axis should be simplified, and specify only the year

222

Figure 6 requires the correction of the text adjustment to the µm scale bars.

Scientific Soundness: is the study correctly designed and technically sound? Are the analyses performed with the highest technical standards? Are the data robust enough to draw the conclusions? Are the methods, tools, software, and reagents described with sufficient details to allow another researcher to reproduce the results?

The paper is scientifically sound and will be of interest to the broad audience as it is reporting results of the demanding long term environmental monitoring. Researchers have performed hydrochemical, hydrobiological and ichthyological studies and kept accurate records of the environmental changes of the heavily polluted arctic lake over the past 27 years. Similar studies can be barely found in the open literature and being performed to the highest technical standards the current study will draw the attention of the broad audience. The study has used standardized and widely accepted research methods and thus is fully reproducible.

However, it is necessary to make a few comments

22, 488 of tolerant and adapted species.

If we take these terms as synonyms, which is possible in a certain context, then we have a tautology. In general, "tolerant" is a momentary characteristic of stability, while "adapted" implies the presence of a previous period of the adaptation process. If the authors emphasize the acquired adaptation, then only the term “adapted” should be left.

147

At this point, it was expected that a description of the methods of statistical calculations, including methods for estimating the average (which is important for characteristics that do not have normal distribution), methods for comparing averages, methods for estimating regression coefficients and their significance would be given. Since in the text the significance levels for differences in statistical parameters are often given, it is necessary to describe the methodology for such calculations.

175-176 uniform distribution of most elements

The specific term “uniform distribution”is used in probabilistic statistics, and in given context immediately throws off a smooth narrative, since the distribution of concentrations of any elements in water even in one horizon cannot be equivalent, but always – lognormal, but varies along the horizons. The authors’idea is clear, but the word order should be changed to eliminate the false impression, for example, “close average levels of concentrations in different parts of the water area”.

233, 234 (Microcystis pulverea f. delicatissima W. & GSWest) Elenk., 1938)

The bracket is missing

244 the maximum values of seasonal phytoplankton biomass also increased up to 10.68 g/m3 (Fig. 7a).

In addition to this outlier, it is clearly visible that the sample volume in 2011-2016 increased sharply, i.e. such outliers may not have been observed in previous years simply because of the small number of samples. Here a correct statistical estimate of the significance of the observed differences should be given; it seems that the significance level will be low.

246-251

Due to the lack of a description of the statistical calculations methodology, the content of Table 3 remains unclear.

  1. Apparently, the average values are given, but why are they chosen, because the estimates of biomass and plankton abundance always do not follow the normal distribution, and median or geometrical mean is more indicative?

  2. What characterizes the given ranges - the variability of indicators in samples or the variation of average annual values?

  3. What statistical criteria are used to make a conclusion about the growth of different parameters? Due to the absence of a normal distribution, T-test cannot be used, but other (non-parametrical) criteria are not described, or are these conclusions not proven statistically at all?

384 - 419

384 The average size and weight indices of the sparsely rakered whitefish individuals in Lake Kuetsjarvi were always higher than those of the densely rakered whitefish individuals (p < 0.001) (Table 6).

Average body sizes calculated on the entire sample do not make biological sense, since they are determined, primarily, by the age composition of the sample. As is seen in Fig.9, the length and weight distributions differ by year and, therefore, the average estimates will shift depending on the representation of individuals of different age in samples. Moreover, as the distribution of fish of different age cannot and does not have normal distribution, the use of parametric criteria for comparing the average values is unacceptable, however, in the methodology there are no any indications on used criteria at all. It remains unclear how this significance level of differences was determined? It is necessary to complete the methodological section and perform a correct comparison. Without this, all conclusions about the difference in body sizes remain unproven.

Why did the authors not follow along the usual method for ichthyology – comparing the morphological characteristics of fish of the same age, which can be done using parametric criteria? Correct conclusions about the difference in size and weight characteristics can be obtained either after unification of the compared samples by age or (if the age is unknown) in the analysis of “fatness”, the study of the regression of body mass along its length. Comparable data must be provided. All differences in average fish size and mass characteristics in different periods cannot be interpreted due to environmental reasons.

409 Thus, here one can also distinguish between smaller and larger individuals ...(Fig. 9b, c, d, e, f).

All frequency comparisons of shown in Fig.9 should be accompanied by an assessment of the significance of the differences. This is not in the text, so practically no phrase has been proved. If it is important for the authors to show changes in the proportions of large and small individuals, they could divide the entire range of values into 2-3 intervals, for which calculate the frequency and perform simple comparisons using statistical criteria. The authors’ logic is clear, they may be right in their preliminary conclusions, but the evidence base for this is also necessary. Statistical criteria should be used to make conclusions about the frequency differences.

Interest to the Readers: Are the conclusions interesting for the readership of the Journal? Will the paper attract a wide readership, or be of interest only to a limited number of people? (please see the Aims and Scope of the journal).

I have enjoyed reading the manuscript and have no doubts that it will attract a large number of readers dealing with environmental conservation, protection, pollution prevention, assessment of environmental impacts and risks, analysis and monitoring of biodiversity, environmental economics, the policy as well as readers working with the development and application of environmental information tools and decision support systems. The paper fully fits the aims and scope of the journal. The long term monitoring is demanding in terms of both financial resources and thorough planning and thus not many of such studies are performed. The study area is unique in terms of the levels of industrial pollution and thus such data are even harder to obtain.

Overall Merit: Is there an overall benefit to publishing this work? Does the work provide an advance towards the current knowledge? Do the authors have addressed an important long-standing question with smart experiments?

The study of Zubova et al. provides new insights into the environmental adaptation strategies of the aquatic species to the heavy metal pollution in the arctic. The growing global demand for the metals and depletion of the existing mines are driving new exploration activities towards the circumpolar areas. The data of Zubova et al. are not only of interest to a broad range of environmental scientists but will help policymakers to develop rational-legal frames for the expanding metallurgical industry.

English Level: Is the English language appropriate and understandable?
The paper is written with understandable English, but I would recommend the correction by a native speaker.

Author Response

Dear Reviewer, thank you very much for the enormous and attentive work you have done on our article!
Thank you for your valuable comments!
A good review always develops a researcher, makes him think and look at his results from another perspective.

We tried to answer all your comments.

- "Of tolerant and adapted species. If we take these terms as synonyms, which is possible in a certain context, then we have a tautology. In general, "tolerant" is a momentary characteristic of stability, while "adapted" implies the presence of a previous period of the adaptation process. If the authors emphasize the acquired adaptation, then only the term “adapted” should be left".

We agree with the remark. The text left only the term "adaptive".

"At this point, it was expected that a description of the methods of statistical calculations, including methods for estimating the average (which is important for characteristics that do not have normal distribution), methods for comparing averages, methods for estimating regression coefficients and their significance would be given. Since in the text the significance levels for differences in statistical parameters are often given, it is necessary to describe the methodology for such calculations».

We agree with the remark. Made additions to the "Materials and methods"section.

- "The signature to the x-axis should be simplified, and specify only the year".

Fixed.

- "The specific term “uniform distribution”is used in probabilistic statistics, and in given context immediately throws off a smooth narrative, since the distribution of concentrations of any elements in water even in one horizon cannot be equivalent, but always – lognormal, but varies along the horizons. The authors’idea is clear, but the word order should be changed to eliminate the false impression, for example, “close average levels of concentrations in different parts of the water area".

Big apologies, here is an inaccurate translation from Russian into English.Сorrect on: "In a study by the Hydrochemical Institute conducted in 1969-1971, close average levels of concentrations of most elements was found in Lake Kuetsjarvi in different parts of the water area and water horizons [45]". Changes made to the text. 

- "Figure 6 requires the correction of the text adjustment to the µm scale bars".

Fixed.

- "(Microcystis pulverea f. delicatissima W. & GSWest) Elenk., 1938).

The bracket is missing".

Fixed.

- "In addition to this outlier, it is clearly visible that the sample volume in 2011-2016 increased sharply, i.e. such outliers may not have been observed in previous years simply because of the small number of samples. Here a correct statistical estimate of the significance of the observed differences should be given; it seems that the significance level will be low". 

- "Due to the lack of a description of the statistical calculations methodology, the content of Table 3 remains unclear.

  1. Apparently, the average values are given, but why are they chosen, because the estimates of biomass and plankton abundance always do not follow the normal distribution, and median or geometrical mean is more indicative?
  2. What characterizes the given ranges - the variability of indicators in samples or the variation of average annual values?
  3. What statistical criteria are used to make a conclusion about the growth of different parameters? Due to the absence of a normal distribution, T-test cannot be used, but other (non-parametrical) criteria are not described, or are these conclusions not proven statistically at all?"

Thank you for the valuable comments that we tried to take into account. Statistical significance of the period’s differences was not evaluated. The periods of the lake ecosystem development based on plankton indicators were distinguished according to several criteria: 1 – change in the species composition – large taxonomic categories proportion; 2 – change of the dominant species; 3 – change in median abundance along with the growth of cases with extremely high values. Of course, highlighting the periods is artificial measure that allows to describe the results more systematically. Authors do not pretend for statistically proven of the selected periods, and in table 3 there are the plankton indicators in different research periods. We changed the averages values to a more revealing median.

- "The average size and weight indices of the sparsely rakered whitefish individuals in LakeKuetsjarvi were always higher than those of the densely rakered whitefish individuals (p < 0.001)(Table 6).

Average body sizes calculated on the entire sample do not make biological sense, since they are determined, primarily, by the age composition of the sample. As is seen in Fig.9, the length and weight distributions differ by year and, therefore, the average estimates will shift depending on the representation of individuals of different age in samples. Moreover, as the distribution of fish of different age cannot and does not have normal distribution, the use of parametric criteria for comparing the average values is unacceptable, however, in the methodology there are no any indications on used criteria at all. It remains unclear how this significance level of differences was determined? It is necessary to complete the methodological section and perform a correct comparison. Without this, all conclusions about the difference in body sizes remain unproven.

Why did the authors not follow along the usual method for ichthyology – comparing the morphological characteristics of fish of the same age, which can be done using parametric criteria? Correct conclusions about the difference in size and weight characteristics can be obtained either after unification of the compared samples by age or (if the age is unknown) in the analysis of “fatness”, the study of the regression of body mass along its length. Comparable data must be provided. All differences in average fish size and mass characteristics in different periods cannot be interpreted due to environmental reasons.

Thus, here one can also distinguish between smaller and larger individuals ...(Fig. 9b, c, d, e, f).

All frequency comparisons of shown in Fig.9 should be accompanied by an assessment of the significance of the differences. This is not in the text, so practically no phrase has been proved. If it is important for the authors to show changes in the proportions of large and small individuals, they could divide the entire range of values into 2-3 intervals, for which calculate the frequency and perform simple comparisons using statistical criteria. The authors’ logic is clear, they may be right in their preliminary conclusions, but the evidence base for this is also necessary. Statistical criteria should be used to make conclusions about the frequency differences".

Thank you for the valuable comments that we tried to take into account. We have made significant changes to the "Results and Discussion" section. Check them out, please. At this stage of the study, data compilation, we did everything we could.

Comparison of the parameters of the dependence of mass on length will not give us anything, since these parameters change due to changes in both the physiological state of the fish (fat content, fattening, maturity of the gonads), and body shape. It is clear that with the same length, the mass of the chasing individual will be less than the mass of the tall one, but we are unlikely to be able to say with certainty that at different times the body shape of the fish did not change.

With best regards,
Zubova Elena